# Automatic ECG Classification Using Continuous Wavelet Transform and Convolutional Neural Network

**DOI:** 10.3390/e23010119

**Published:** 2021-01-18

**Authors:** Tao Wang, Changhua Lu, Yining Sun, Mei Yang, Chun Liu, Chunsheng Ou

**Affiliations:** 1School of Computer and Information, Hefei University of Technology, Hefei 230009, China; lch6208@163.com; 2Institute of Intelligent Machines, Chinese Academy of Sciences, Hefei 230031, China; ynsun@iim.ac.cn; 3Beijing Huaru Technology Co., Ltd., Hefei Branch, Hefei 230088, China; ym1377689441@hotmail.com; 4School of Electrical Engineering and Automation, Hefei University of Technology, Hefei 230009, China; dqlch03@hfut.edu.cn

**Keywords:** arrhythmia, continuous wavelet transform, convolutional neural network, deep learning, ECG classification, heartbeat classification

## Abstract

Early detection of arrhythmia and effective treatment can prevent deaths caused by cardiovascular disease (CVD). In clinical practice, the diagnosis is made by checking the electrocardiogram (ECG) beat-by-beat, but this is usually time-consuming and laborious. In the paper, we propose an automatic ECG classification method based on Continuous Wavelet Transform (CWT) and Convolutional Neural Network (CNN). CWT is used to decompose ECG signals to obtain different time-frequency components, and CNN is used to extract features from the 2D-scalogram composed of the above time-frequency components. Considering the surrounding R peak interval (also called RR interval) is also useful for the diagnosis of arrhythmia, four RR interval features are extracted and combined with the CNN features to input into a fully connected layer for ECG classification. By testing in the MIT-BIH arrhythmia database, our method achieves an overall performance of 70.75%, 67.47%, 68.76%, and 98.74% for positive predictive value, sensitivity, F1-score, and accuracy, respectively. Compared with existing methods, the overall F1-score of our method is increased by 4.75~16.85%. Because our method is simple and highly accurate, it can potentially be used as a clinical auxiliary diagnostic tool.

## 1. Introduction

Arrhythmia refers to irregular heart rhythm and is one of the main causes of cardiovascular disease (CVD) death. Most arrhythmias are not serious, but some are harmful or even life-threatening [1]. For example, atrial fibrillation can lead to strokes and cardiac arrest. It is very dangerous and needs to be treated immediately. According to the World Health Organization (WHO) report, CVD caused approximately 17.5 million deaths in 2012, accounting for 30% of global deaths [2]. By 2030, the number of CVD deaths is expected to increase to 23 million. Furthermore, the cost of CVD-related treatments, including medication, is very expensive. It is estimated that the cost in low- and middle-income countries is approximately US $3.8 trillion from 2011 to 2025 [2].

Early detection and effective treatments such as vagal maneuvers and medications can prevent the occurrence of CVD. In the clinical setting, the arrhythmia is usually diagnosed by analyzing the heartbeat of an electrocardiogram (ECG) signal. An ECG signal consists of a series of heartbeats (or called waves) that repeat periodically in time and represents the electrical activity of the heart over time [3]. The doctor checks these heartbeats to diagnose the presence of arrhythmia, while the process is time-consuming and labor-intensive.

To this end, researchers have developed a method to automatically classify heartbeats in ECG signals (also known as heartbeat classification in some papers) [1,2,4,5]. Most methods consist of feature extraction and classification. The heartbeat morphological and RR interval features are usually used. For classification, different algorithms have been used, including artificial neural networks (ANNs), support vector machines (SVMs), multi-view-based learning, and linear discriminants (LDs) [6]. Despite the good performance achieved by these methods, the ECG waves and their morphological characteristics of different patients have significant variations, and for the same patient, the ECG waves at different times are also different. The fixed features used in these methods are not enough to accurately distinguish arrhythmia of different patients. Recently, with the rapid development of deep neural networks, deep learning-based methods have attracted more and more attention. Deep learning, as a representation learning method, can automatically extract discriminant features from the training data. Several studies [6,7,8] show that deep learning-based methods can extract more abstract features and resolve variations between patients in ECG classification.

However, there is another problem in the ECG classification. The ECG signal is usually composed of different frequency components and even noises, which increases the difficulty of deep learning-based method to extract discriminant features. A naturally conceivable way is to transform the ECG signal to time-frequency domain to avoid the effects of aliasing of different frequencies components. There are two widely used time-frequency techniques: Wavelet Transform (WT) [9] and Short-Time Fourier Transform (STFT) [10]. WT inherits and develops the idea of STFT, but unlike STFT, WT can not only provide high-frequency resolution and low time resolution at low frequencies, but also have high time resolution and low-frequency resolution at high frequencies [11]. Generally, WT can obtain better time-frequency domain analysis results than STFT.

Motivated by these challenges, we develop an automatic ECG classification method based on Continuous Wavelet Transform (CWT) and Convolutional Neural Network (CNN) for ECG classification, where CWT refers to WT using continuous wavelet function. CNN is a deep learning method that imitates the human visual system, which has been successfully used for image classification and video recognition [12,13]. The CWT is used to transform the ECG heartbeat signal to the time-frequency domain and CNN is used to extract features from the 2D scalogram composed by the above-decomposed time-frequency components. The method has combined the capabilities of CWT in multi-dimensional signal processing and CNN in image feature extraction. Arrhythmia usually not only affects the shape of the heartbeat, but also changes the surrounding RR intervals. To makes full use of all information for ECG classification, the RR interval features are also extracted and fused into our CNN. By testing in the MIT-BIH arrhythmia database [14], our method achieves an overall performance of 70.75%, 67.47%, 68.76%, and 98.74% for positive predictive value, sensitivity, F1-score, and accuracy, respectively. Compared with existing methods [5,15,16,17,18], the overall F1-score of our method is increased by 4.75~16.85%.

The paper is organized as follows. Section 2 introduces the current classification paradigms and existing methods. The proposed ECG classification method based on CWT and CNN is presented in Section 3. The results and discussion are described in Section 4 and Section 5. Section 6 concludes the paper.

## 2. Literature Review

### 2.1. Classification Paradigms

Over the past two decades, many automatic ECG classification methods have been proposed. These works can be grouped into three classification paradigms: intra-patient paradigm, inter-patient paradigm, and patient-specific paradigm [1]. The intra-patient paradigm divides the dataset into training and test subsets based on heartbeat labels [1], so an ECG recording will appear in two subsets. According to research by de Chazal et al. [4], the intra-patient paradigm would cause the model to learn the patient’s characteristics during the training phase and shows almost 100% classification accuracy during the test phase. However, a well-trained model must deal with the heartbeat of unseen patients [7].

In order to be consistent with the practical situation, de Chazal et al. [4] present an inter-patient paradigm. In the inter-patient paradigm, the training and test subsets consist of different ECG recordings [1], so the inter-variation of patients must be considered when building the model. Although the performance of the inter-patient paradigm model is worse than that of the intra-patient paradigm model, it has a better generalization ability and is consistent with clinical practice. The patient-specific paradigm is a hybrid of the inter-patient paradigm and the intra-patient paradigm [1], also proposed by de Chazal et al. [19]. In this paradigm, first use the inter-patient paradigm to train the global model, and then use part of the new patient data to fine-tune the model to form a patient-specific model. Compared with the inter-patient paradigm, the patient-specific paradigm can achieve better performance. However, this method requires a doctor to label part of the data for each new patient and an engineer to fine-tune the model, which leads to the limitation of clinical application. In the paper, we focus on the performance of our method in the inter-patient paradigm.

### 2.2. Existing Methods

Due to the overfitting of the intra-patient paradigm, we only introduce the inter-patient paradigm and patient-specific paradigm methods. For the inter-patient paradigm, de Chazal et al. [4] develop a linear discriminant method for ECG classification. In the method, they adopt morphological and dynamic features to represent the heartbeat, and use maximum likelihood estimation to determine its parameters. Their model obtains a sensitivity of 75.9% and a positive predictive value of 38.5% for SVEB. Ye et al. [5] apply independent component analysis (ICA) and wavelet transform to extract morphological features, and combine the heartbeat interval to represent the heartbeat. An SVM classifier, then, is used to classify the heartbeat. The overall sensitivity and positive predictive value of the model are 53.46% and 62.79%, respectively. Chen et al. [16] use a random projection matrix to derive projected features, and propose a new ECG heartbeat classification method based on the projected and dynamic features. A sensitivity of 49.69% and a positive predictive value of 54.77% is achieved.

For the patient-specific paradigm, Hu et al. [20] present a mixture-of-experts (MOE) method for ECG classification. In the method, each heartbeat contains 14 sample points around its R peak. First, train the global classifier based on the ECG recordings of different patients. Then, use the five-minute doctor-annotated ECG signal of the new patient to train the local classifier. Finally, combined the global classifier and local classifier to form a patient-specific classifier. Their method achieves a sensitivity of 82.6% and a positive predictive value of 77.7%. Ince et al. [21] develop a patient-specific neural network (NN) using representative beats randomly selected from training records and the first 5 min of the heartbeat of the new patients. The average accuracy-sensitivity performances of the model for VEB and SVEB are 98.3–84.6% and 97.4–63.5%, respectively. Ye et al. [22] proposed a patient-specific model based on their early work [5]. In the work, they used partial heartbeat data from new patients to fine-tune the model in [5], getting an average classification accuracy of 99.4% for VEB and 98.3% for SVEB.

## 3. Methods

The method consists of preprocessing, feature extraction, and classification, as shown in Figure 1. The preprocessing contains ECG signal denoising, heartbeat segmentation and RR Interval extraction. Below we introduce each part.

### 3.1. Dataset

The well-known MIT-BIH arrhythmia database [14] is used as the benchmark dataset to evaluate the proposed method in the paper. The database contains 48 half-h recordings obtained from 47 subjects. Twenty-three recordings are randomly selected from the 4000 24-h ambulatory ECG recordings and intended as a representative sample of routine clinical. The remaining 25 recordings are also obtained from the above ambulatory ECG recordings, but contain rare but clinically significant arrhythmias. Each recording consists of two leads (i.e., lead A and B), 360 samples per second with an 11-bit resolution over the 10-mV range ECG signals. The lead A is the modified-lead II (ML II) and the lead B is lead VI, VII, V2, V4, or V5, depending on the recording. As ML II is ubiquitous in the above records, we use ML II for ECG classification in the paper. These recordings are individually labeled by two or more cardiologists and divided into 15 arrhythmia types. By following the recommendations of the Association for the Advancement of Medical Instrumentation (AAMI), we further group these arrhythmias into five classes, as shown in Table 1, and four records (i.e., 102, 104, 107, and 217) that have paced beats are removed. Furthermore, since the Q class is practically nonexistent, we ignore it like others [23,24].

As we mentioned above, in the paper we focus on the performance of our method in the inter-patient paradigm. To facilitate direct comparison with existing works, a widely used data division method proposed by de Chazal et al. [4] is employed to split the database. The MIT-BIH arrhythmia database is split into DS1 and DS2 datasets [4], each of which is composed of 22 records that have a similar proportion of beat types. The first dataset is used for training, and the second is used to test the performance of the method. None of the patients exists in two datasets.

### 3.2. ECG Preprocessing and Heartbeat Segmentation

The clinically collected ECG signal is normally corrupted by various noises such as baseline wandering, electromyography disturbance, and power line interference, which makes it difficult to extract useful information from the raw ECG signal. Therefore, a filtering step is required before further processing. As excessive filtering would lead to the loss of useful information, we only remove the noise—baseline wandering, which has an important impact on ECG classification [24]. The baseline wandering is caused by respiration or patient movement [1]. Following previous works [25], two median filters (i.e., a 200 ms width median filter and a 600 ms width median filter) are adopted to achieve the baseline wandering, and then subtract it from the raw signals to yield the baseline-corrected ECG signal. Figure 2a,b shows the effect of baseline wandering removal. Compared with other filter techniques such as regular infinite impulse response (IIR) and finite impulse response (FIR), the median filter can eliminate outliers neatly without increasing phase distortion.

Before the ECG classification, we need to segment the individual heartbeats from the ECG signal. This usually requires accurate detection of QRS waves and fiducial points of heartbeats. However, this is not the goal of the paper, and at present, there are many high-precision (>99%) QRS waves and fiducial points positioning methods been developed in the literature. In the paper, we take the annotated R-peak location as the fiducial point and segment the ECG signal into a series of heartbeats. This can allow us to directly compare the performance with other works. For each heartbeat, we obtain a fixed-size of 200 samples ECG signal by taking 90 samples before and 110 samples after the R-peak. These sample points have captured the most important waves of heartbeats. An illustration of the segmentation is shown in Figure 2c.

### 3.3. Time-Frequency Scalogram via CWT

As the ECG signal is composed of different frequency components, in this study we transform the ECG signal to the time-frequency domain to facilitate feature extraction. CWT is the most commonly used time-frequency analysis tool, which uses a family of wavelet functions to decompose a signal in the time-frequency domain. It inherits and develops the localization idea of STFT, but unlike STFT, CWT can provide high time resolution and low-frequency resolution in the high frequencies, and high-frequency resolution and low time resolution in the low frequencies by adjusting the scale and translation parameters [11]. Formally, given a signal x(t), the CWT is defined as
(1)$$C_{a}\left(b\right)=\frac{1}{\sqrt{a}}\int_{-\infty }^{\infty }x\left(t\right)\cdot \varphi \left(\frac{t-b}{a}\right)dt$$
where *a* is a scale parameter, *b* is a translation parameter, and $\varphi \left(t\right)$ is the wavelet function (also known as mother wavelet). The scale can be converted to frequency by
(2)$$F=\frac{F_{c}*f_{s}}{a}$$
where $F_{c}$ is the center frequency of the mother wavelet, $f_{s}$ is the sampling frequency of signal x(t) [26].

Among them, the choice of mother wavelet is often critical to the effect of time-frequency analysis. In the paper, the Mexican hat wavelet (mexh) is taken as the mother wavelet as it is close to the shape of the QRS waves and widely used in ECG signal analysis, which is defined as
(3)$$\varphi \left(t\right)=\frac{2}{\sqrt{3}\sqrt[{4}]{\pi }}\exp\left(-\frac{t^{2}}{2}\right)\left(1-t^{2}\right)$$
By using different scale factors of CWT, the wavelet coefficients of the signal at different scales are obtained. These wavelet coefficients can be regarded as a 2D scalogram of ECG signal in the time-frequency domain.

Figure 3 shows the time-domain ECG heartbeat signal and scalogram of normal heartbeat and premature ventricular contraction (PVC) heartbeat. Both signals have 200 sampling points and are sampled at a frequency of 360 Hz, decomposed by the Mexican hat wavelet. It can be seen from the scalogram that the PVC heartbeat is obviously different from the normal heartbeat. This indicates that it is possible to carry out heartbeat classification using the scalogram. However, it is difficult to explicitly build the relationship between scalogram and abnormal conditions. To solve the problem, CNN is developed to automatically extract the potential relationship between different arrhythmias and normal heartbeats in the paper.

### 3.4. ECG Classification Based on CNN

CNN is a deep learning that uses convolution operations to replace general multiplication in deep neural networks. It can automatically extract discriminant features through the training process and has been widely used in classification tasks in recent years, especially in image recognition [27]. CNN’s outstanding achievements are attributed to two important concepts: sparse interaction and parameter sharing [28]. Sparse interaction is achieved by making the size of the convolution kernel much smaller than the input. It reduces the computational complexity of the model and improves its statistical efficiency. Parameter sharing refers to using the same parameters in the multiplication operation, that is, the parameters of each convolution kernel are the same when processing different positions of the input.

A typical CNN has multiple layers, the most important of which is the convolution layer. The convolution layer applies a set of weights called filters or kernels to extract features. Basically, the relevant high-level features can be extracted by increasing the number of convolution layers. The backpropagation (BP) error algorithm is used to train the weights of the convolution kernel. Other layers that are commonly used in CNN are rectified learning units (ReLU) layer, batch normalization layer, and pooling layer. ReLU layer is used as an activation function to achieve nonlinear capabilities. Batch normalization layer is usually placed between the convolution layer and the ReLU layer. This layer normalizes the feature map of each channel, reducing training time and sensitivity of network initialization. The pooling layer, also known as the subsampling layer, is used to reduce the feature dimension and speed up the training process. The action of this layer is to calculate the average or maximum convolution features within adjacent neurons placed in the previous convolution layer [26]. The last layer of CNN is normally connected to one or more fully connected neurons that use to compute the class scores. In this study, we use the CNN shown in Figure 4. To achieve a clear representation, the convolution layer, batch normalization layer, and ReLU layer are visually combined into a convolution unit. Three consecutive convolution and pooling operations are used to extract features from the scalogram. After the last operation, a 64-dimensional feature is obtained. Noted that the size of the original scalogram is 100 × 200, we resampled it to 100 × 100 to reduce the computational cost. It should be pointed out that the main information of the ECG signal is concentrated in 0~50 Hz, so the resampling will not cause performance degradation.

Arrhythmia usually not only affects the shape of the heartbeat, but also changes the surrounding RR intervals (also known as R peak intervals). Therefore, we also combine the RR interval information into our CNN for ECG classification. Four widely used RR intervals features (e.g., previous-RR, post-RR, ratio-RR, and local-RR) are extracted. The previous-RR is the RR interval between the current heartbeat and the previous heartbeat [17]. The post-RR is the RR-interval between the current heartbeat and the following heartbeat [17]. The ratio-RR is the ratio of previous-RR and post-RR. The local-RR is determined by the average of ten previous RR-intervals of the current heartbeat. The previous-RR, post-RR, and local-RR have subtracted the average RR-interval to eliminate the inter-patient variation. The fusion features are input into two fully connected layers for classification. The details of our CNN are listed in Table 2.

### 3.5. Training Setting

The cross-entropy is taken as the loss function, and Adam is used as the optimizer. Compared with other optimizers, Adam can usually speed up network training. The weights of the convolution layers and fully connected layers are initialized using He initialization [29]. The learning rate is 0.001, which is reduced by 0.1 times every 5 epochs. The batch size of the model is 1024 and the maximum epoch is set to 30. CNN is implemented using PyTorch [30] and trained on the NVIDIA GeForce RTX 2080Ti graphical processing unit. The code is located at https://github.com/JackAndCole/ECG-Classification-Using-CNN-and-CWT.

## 4. Results

To evaluate the performance of the method, three widely used metrics, positive predictive value (PPV), sensitivity (SE), and accuracy (ACC), are adopted, which are defined as
(4)$$PPV_{i}=\frac{TP_{i}}{TP_{i}+FP_{i}}$$
(5)$$SE_{i}=\frac{TP_{i}}{TP_{i}+FN_{i}}$$
(6)$$ACC_{i}=\frac{TP_{i}+TN_{i}}{TP_{i}+TN_{i}+FP_{i}+FN_{i}}$$
where $TP_{i}$ (true positive) and $FN_{i}$ (false negative) refer to the number of the *i*-th class correctly predicted and the number of the *i*-th class classified into other classes, respectively. $TN_{i}$ (true negative) and $FP_{i}$ (false positive) are the number of other classes that is not classified as the *i*-th class and the number of other classes is predicted as the *i*-th class, respectively.

As the heartbeat types are imbalanced, the F1-score is also used as the performance metric in the paper, it is defined as
(7)$$F1_{i}=\frac{2\cdot PPV_{i}\cdot SE_{i}}{PPV_{i}+SE_{i}}$$
It takes into account both the positive predictive value and the sensitivity, and is often used as an overall performance metric for comparing multiple methods. In the imbalanced data, it is usually more useful than accuracy [31].

According to the recommendations of AAMI, we group the heartbeats of the MIT-BIH arrhythmia database and use the method of de Chazal et al. [4] to divide the database into DS1 and DS2 datasets. DS1 is used for training and DS2 is used to test the method. In order to make a fair comparison, in this study we only compare it with the method that using the same strategy. As the SVEB and VEB classes are more important than other classes in the ECG classification, we analyze these two classes in detail, as shown in Table 3. Apart from the sensitivity of SVEB, our method achieves the best performance among all metrics in SVEB and VEB. For example, compared to Ye et al., the second best F1-score in SVEB, the positive predictive value, sensitivity, F1-score, and accuracy of our method in SVEB has improved by 37.20%, 13.55%, 25.02%, and 2.47%, respectively.

The performance of other classes is listed in Table 4. Except for F class, our method has achieved better or comparable performance in other classes. Especially, the overall F1-score of our method is improved by 4.57–16.85%. For the F class, it is mainly composed of the fusion of ventricular and normal beat, which is very close to the normal heartbeat, making it difficult for the algorithm to classify them. Existing methods usually predict a large number of N as F class or F as N class. In our method, CNN can automatically extract the discriminant features, but the number of F class is limited, which makes our method presents the same situation (see the confusion matrix of our method, Table 5) as described above. Zhang et al., although, achieve the best performance in F class, a large number of N class is misclassified as F class.

## 5. Discussion

A CWT-based CNN is proposed for ECG classification in this study. The method makes use of the feature representation capabilities of CNN. In order to verify this capability, we analyze it in this part. In addition, the wavelet function is very important for CWT. We also discuss the impact of different wavelet types on performance here.

**CNN feature visualization:** As a representation learning method, our CNN-based methods can automatically extract discriminant features from the data. To verify the capability, we use t-distributed stochastic neighbor embedding (t-SNE) to visualize the extracted CNN features [32]. t-SNE is a nonlinear dimensionality reduction method that is widely used in deep learning to visualize high-dimensional data in two-dimensional or three-dimensional space. Figure 5 shows the t-SNEs of the raw scalogram and representation features obtained from three convolution units. To achieve a good visualization, the number of samples in the figures has been reduced. From Figure 5, we can see that there are some outliers in the raw scalogram, and different types of heartbeats are mixed together. While in the convolution units, as the layer deepens, the outliers gradually decrease and the heartbeat gathers. Especially in the last convolution unit, there is an obvious clustering of different heartbeats. This means that our CNN can effectively extract features, and as the network deepens, the extracted features become more and more discriminative.

**Impact of wavelet types:** CWT is the most commonly used signal analysis tool in the time-frequency domain, but there is no uniform standard for the selection of wavelet functions. In the paper, besides mexh, we also analyze three wavelet functions that widely used for ECG signals, namely morl, gaus8, gaus4. The overall performance of four wavelet functions is listed in Table 6. Mexh achieves the best performance in terms of positive predictive value, F1-score, and accuracy, except that slightly lower than gaus4 in sensitivity. To explore the reasons behind it, we plot these wavelet functions, as shown in Figure 6. Among these wavelet functions, mexh is the closest to the waveform of the ECG signal, followed by gaus4. Gaus8 and morl are different from the waveform of the ECG signal. It is interesting to find that the higher the similarity between the waveform of the ECG signal and the wavelet function, the better the performance. Therefore, it is recommended to use a wavelet function close to the signal when performing signal analysis like ECG signals.

## 6. Conclusions

We developed a novel ECG classification method based on CWT and CNN in the paper. To avoid the effects of aliasing of different frequency components, CWT is first used to transform the ECG heartbeat signal into the time-frequency domain. Then, CNN is used to extract features from a scalogram composed by decomposed time-frequency components. The method can make full use of the advantages of CWT in multi-dimensional signal processing and CNN in image recognition. By testing it on the MIT-BIH arrhythmia database using the inter-patient paradigm, an overall performance of 70.75%, 67.47%, 68.76%, and 98.74% for the positive predictive value, sensitivity, F1-score, and accuracy is achieved. Due to the highly accurate ECG classification, our method can potentially be used as a clinical auxiliary diagnostic tool. In general, arrhythmia, as one of the main causes of cardiovascular disease, is necessary to diagnose it at an early stage. Upon a proper early diagnosis, effective treatment like vagal maneuvers or medications can reduce arrhythmia and avoid cardiovascular disease.

Although good overall performance achieved by our method, the performance of the F class still needs to be improved. As we discussed above, this is because the F class is mainly composed of the fusion of ventricular and normal beat, and the number of F class is significantly less than other classes. In general, this can be improved by adding more annotated ECG data. However, labeling ECG heartbeats are very expensive and time-consuming. Nowadays, there are many publicly available unlabeled ECG databases, and the use of unsupervised learning such as autoencoder may further improve the performance of the F class in an inexpensive way. In the future, we will try to carry out related work.

## Figures and Tables

**Figure 1 entropy-23-00119-f001:**
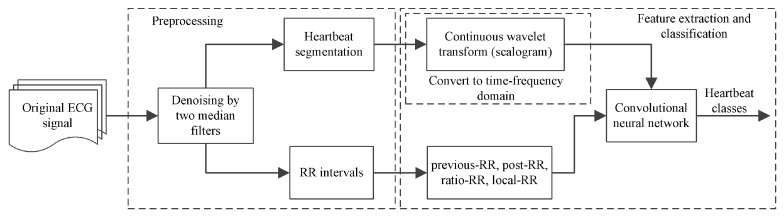
Flowchart of our proposed method.

**Figure 2 entropy-23-00119-f002:**
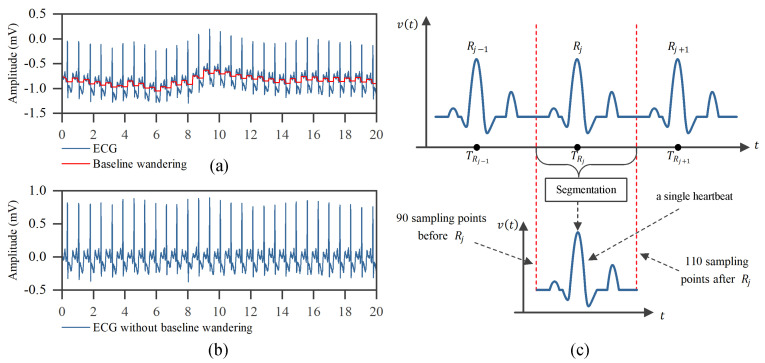
Baseline wandering removal and heartbeat segmentation. Subfigures (**a**,**b**) represent before and after removing the baseline wandering, respectively. Subfigure (**c**) is an illustration of heartbeat segmentation.

**Figure 3 entropy-23-00119-f003:**
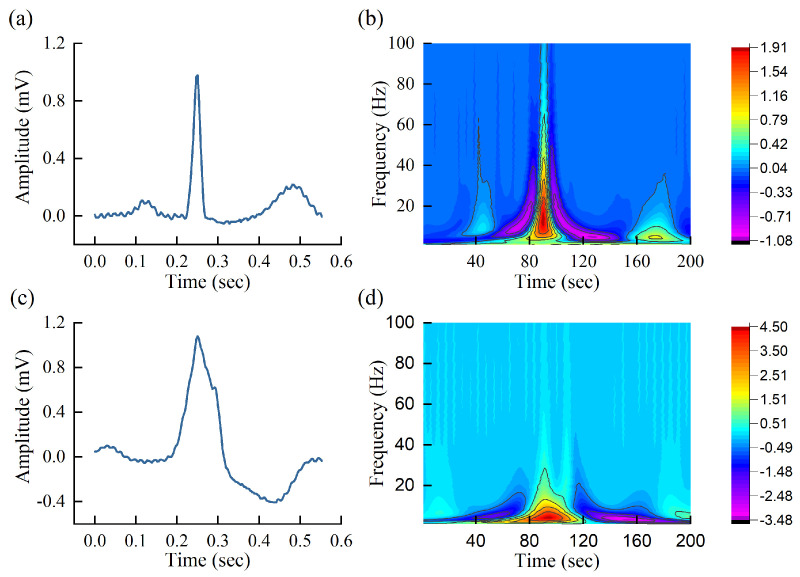
Raw ECG heartbeat signal and CWT scalogram. Subfigures (**a**,**b**) represent the ECG heartbeat signal and CWT scalogram of normal heartbeat, respectively. Subfigures (**c**,**d**) represent the ECG heartbeat signal and CWT scalogram of abnormal heartbeat (premature ventricular contraction (PVC)), respectively.

**Figure 4 entropy-23-00119-f004:**
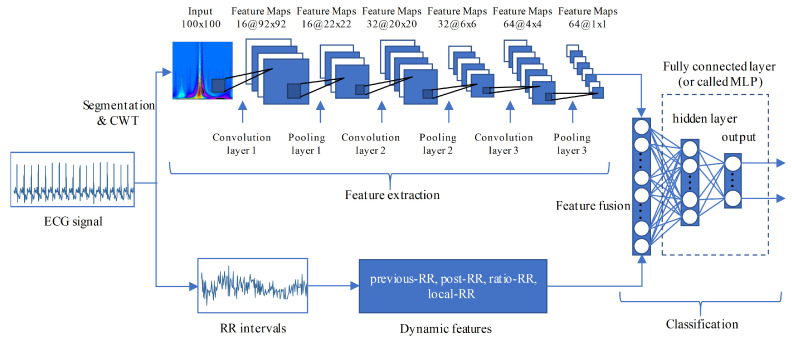
Our CNN architecture. Our Convolutional Neural Network (CNN) not only uses the CWT scalogram of heartbeat, but also adopts the RR interval features.

**Figure 5 entropy-23-00119-f005:**
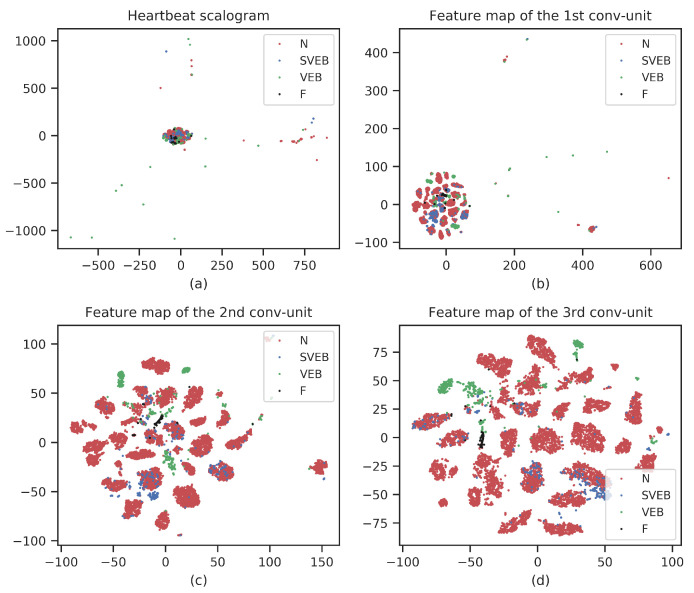
t-SNEs of raw scalogram and representation features obtained from three convolution units.

**Figure 6 entropy-23-00119-f006:**
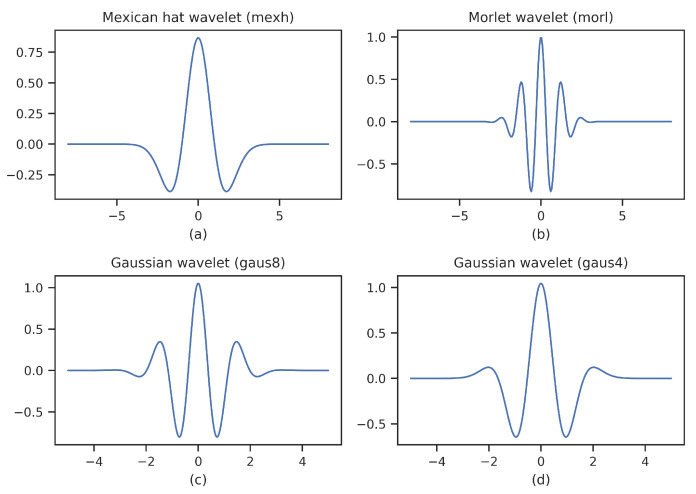
Wavelet functions. Mexh, morl, gaus8, and gaus4 are Mexican hat wavelet, Morlet wavelet, Gaussian wavelet with 8 derivatives, and Gaussian wavelet with 4 derivatives, respectively.

**Table 1 entropy-23-00119-t001:** Mapping of MIT-BIH arrhythmia types and Advancement of Medical Instrumentation (AAMI) classes.

AAMI Classes	Normal (N)	Supraventricular Ectopic Beat (SVEB)	Ventricular Ectopic Beat (VEB)	Fusion Beat (F)	Unknown Beat (Q)
MIT-BIH arrhythmia types	Normal beat (NOR)—N	Atrial premature beat (AP)—A	Ventricular escape beat (VE)—E	Fusion of ventricular and normal beat (fVN)—F	Unclassifiable beat (U)—Q
	Right bundle branch block beat (RBBB)—R	Premature or ectopic supraventricular beat (SP)—S	Premature ventricular contraction (PVC)—V		Fusion of paced and normal beat (fPN)—f
	Left bundle branch block beat (LBBB)—L	Nodal (junctional) premature beat (NP)—J			Paced beat (P)—/
	Atrial escape beat (AE)—e	Aberrated arial premature beat (aAP)—a			
	Nodal (junctional) escape beat (NE)—j				

**Table 2 entropy-23-00119-t002:** The parameters of our CNN architecture.

No.	Layer Name	Kernel Size	Filter	Padding	Stride	Output Shape	Parameters
1	Input1 *	-	-	-	-	100 × 100 × 1	-
2	Conv2D	7 × 7	16	0	1	94 × 94 × 16	784
3	Batch Normalization	-	-	-	-	94 × 94 × 16	64
4	ReLu	-	-	-	-	94 × 94 × 16	-
5	Max pooling	5 × 5	-	0	5	18 × 18 × 16	-
6	Conv2D	3 × 3	32	0	1	16 × 16 × 32	4608
7	Batch Normalization	-	-	-	-	16 × 16 × 32	128
8	ReLu	-	-	-	-	16 × 16 × 32	-
9	Max pooling	3 × 3	-	0	3	5 × 5 × 32	-
10	Conv2D	3 × 3	64	0	1	3 × 3 × 64	18,432
11	Batch Normalization	-	-	-	-	3 × 3 × 64	256
12	ReLu	-	-	-	-	3 × 3 × 64	-
13	Global Max pooling	3 × 3	-	-	-	1 × 1 × 64	-
14	Flatten	-	-	-	-	64	-
15	Input2 **	-	-	-	-	4	-
16	Concatenate	-	-	-	-	68	-
17	Dense	-	-	-	-	32	2208
18	Dense	-	-	-	-	4	132

* is the scalogram of the heartbeat; ** refers to the RR interval features of the heartbeat.

**Table 3 entropy-23-00119-t003:** The classification performance of existing works and our method in SVEB and VEB classes.

Methods	SVEB				VEB			
	Positive Predictive Value	Sensitivity	F1-score	Accuracy	Positive Predictive Value	Sensitivity	F1-score	Accuracy
Liu et al. [15]	39.87%	33.12%	36.18%	95.49%	76.51%	90.2%	82.79%	97.45%
Chen et al. [16]	38.40%	29.50%	33.36%	95.34%	85.25%	70.85%	77.38%	97.32%
Zhang et al. [17]	35.98%	79.06%	49.46%	93.33%	92.75%	85.48%	88.96%	98.63%
Ye et al. [5]	52.34%	61.02%	56.34%	96.27%	61.45%	81.82%	70.19%	95.52%
Garcia et al. [18]	53.00%	62.00%	57.15%	-	59.40%	87.30%	70.70%	-
Our method	89.54%	74.56%	81.37%	98.74%	93.25%	95.65%	94.43%	99.27%

**Table 4 entropy-23-00119-t004:** The classification performance of existing works and our method in all classes.

Classes	Metrics	Methods					
		Liu et al.	Chen et al.	Zhang et al.	Ye et al.	Garcia et al.	Our Method
N	Positive predictive value	96.66%	95.42%	98.98%	97.55%	98.00%	98.17%
	Sensitivity	94.06%	98.42%	88.94%	88.61%	94.00%	99.42%
	F1-score	95.34%	96.90%	93.69%	92.87%	95.96%	98.79%
SVEB	Positive predictive value	39.87%	38.40%	35.98%	52.34%	53.00%	89.54%
	Sensitivity	33.12%	29.50%	79.06%	61.02%	62.00%	74.56%
	F1-score	36.18%	33.36%	49.46%	56.34%	57.15%	81.37%
VEB	Positive predictive value	76.51%	85.25%	92.75%	61.45%	59.40%	93.25%
	Sensitivity	90.20%	70.85%	85.48%	81.82%	87.30%	95.65%
	F1-score	82.79%	77.38%	88.96%	70.19%	70.70%	94.43%
F	Positive predictive value	12.99%	0.00%	13.73%	2.50%	-	2.04%
	Sensitivity	40.72%	0.00%	93.81%	19.69%	-	0.26%
	F1-score	19.70%	0.00%	23.96%	4.43%	-	0.46%
Average	Positive predictive value	56.51%	54.77%	60.36%	53.46%	52.60%	70.75%
	Sensitivity	63.53%	49.69%	86.82%	62.79%	60.83%	67.47%
	F1-score	58.50%	51.91%	64.02%	55.96%	55.95%	68.76%

**Table 5 entropy-23-00119-t005:** The confusion matrix of our proposed CNN.

		Predicted Label				
		N	SVEB	VEB	F	Total
True label	N	43,962	147	79	30	44,218
	SVEB	329	1369	123	15	1836
	VEB	124	13	3079	3	3219
	F	366	0	21	1	388
	Total	44,781	1529	3302	49	49,661

**Table 6 entropy-23-00119-t006:** The overall classification performance of the four wavelet functions.

Mother Wavelet	Positive Predictive Value	Sensitivity	F1-Score	Accuracy
Mexican hat wavelet (mexh)	70.75%	67.47%	68.76%	98.74%
Morlet wavelet (morl)	61.68%	67.13%	63.54%	97.65%
Gaussian wavelet (gaus8)	67.23%	66.97%	65.63%	98.14%
Gaussian wavelet (gaus4)	65.33%	68.18%	66.56%	98.30%

## Data Availability

Publicly available datasets were analyzed in this study. This data can be found here: https://www.physionet.org/content/mitdb/1.0.0/.

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
