# Peer review of "Automatic ECG Classification Using Continuous Wavelet Transform and Convolutional Neural Network"

_entropy, 2021, doi:10.3390/e23010119_

Round 1

Reviewer 1 Report

In this manuscript, the authors describe an ECG classification framework based on continuous wavelet transform (CWT) and convolutional neural network (CNN) for detection of arrhythmia in ECG signals. Time frequency components and time domain features of ECG were used as inputs to the CNN for classification. The authors use positive predictive value, sensitivity, f1-score, and accuracy as metrics for assessing the performance of the proposed classification method. They claim that using the proposed classification framework, they can achieve significantly improved performance compared to existing methods. The methods are well described, and the figures and tables are well written. I have the following suggestions/concerns:

1) In the abstract, the authors refer to an RR interval. It would make the paper more readable if they describe somewhere what an RR interval is for audience who are not in the ECG research field but would still benefit from reading the methods.

2) In the abstract, it would be worthwhile to indicate in someway what the existing methods performance was and how significant the improvement is.

3) The manuscript needs an overall proofreading as the sentences are often not well formed. Some examples are:

i) In the abstract, “Due to the simple yet highly accurate of our method” needs to be formatted.

ii) Line 28: In clinical setting or Clinically

iii) Line 38: been used instead of being studied

iv) Throughout the text, discriminant features instead of discriminate features

v) Line 114: What is MOE method?

vi) Line 301: similarity in place of similar

vii) Line 306: developed in place of develop

viii) “is” is missing

4) The references for MIT-BIH database (line 70) and for existing methods (line 73) are missing.

5) In the existing methods section, it would be nice to include the actual performances for the methods mentioned or refer to the table later.

6) Methods: Why did the authors not consider cross validation for improving performance.

Author Response

Comment 1: In the abstract, the authors refer to an RR interval. It would make the paper more readable if they describe somewhere what an RR interval is for audience who are not in the ECG research field but would still benefit from reading the methods.

Response: We thank the reviewer for pointing out this issue. According to the reviewer's suggestion, in the abstract, we described the RR interval in detail to improve readability.

Comment 2: In the abstract, it would be worthwhile to indicate in someway what the existing methods performance was and how significant the improvement is.

Response: Based on the reviewer’s suggestions, in the abstract we describe the detailed performance improvement of our method compared to existing methods.

Comment 3: The manuscript needs an overall proofreading as the sentences are often not well formed. Some examples are:
i) In the abstract, “Due to the simple yet highly accurate of our method” needs to be formatted.
ii) Line 28: In clinical setting or Clinically
iii) Line 38: been used instead of being studied
iv) Throughout the text, discriminant features instead of discriminate features
v) Line 114: What is MOE method?
vi) Line 301: similarity in place of similar
vii) Line 306: developed in place of develop
viii) “is” is missing

Response: We thank the reviewer for pointing out these grammatical and sentence issues. We have carried out detailed proofreading in the revised manuscript.

Comment 4: The references for MIT-BIH database (line 70) and for existing methods (line 73) are missing.

Response: Based on the reviewer’s suggestion, we added relevant references.

Comment 5: In the existing methods section, it would be nice to include the actual performances for the methods mentioned or refer to the table later.

Response: According to the reviewer's suggestion, in the existing methods section, we added the actual performance of the mentioned methods.

Comment 6: Methods: Why did the authors not consider cross validation for improving performance.

Response: We thank the reviewer for pointing out this issue that needs clarification. In theory, cross-validation can further improve the performance of our method. However, in our research, the combined model of CWT and CNN not only needs to select the parameters of CNN, but also need to determine the parameters of CWT. Due to a large number of parameters in the model and the long cross-validation time, it is almost impossible to select the best parameters for the combined model of CWT and CNN. In fact, in the study, the parameters of CWT and CNN are determined based on trail-and-error and similar related research.

Reviewer 2 Report

Comments
The work presented in the manuscript is aimed at automatic ECG classification based on Continuous Wavelet Transform (CWT) and Convolutional Neural Network (CNN) for the diagnosis of arrhythmia. The disease is the main causes of cardiovascular disease (CVD) death.

The authors propose a machine-learning-based approach for the automatic classification of ECG signals to aid in the diagnosis of arrhythmia. A novel, robust, and accurate method will be invaluable. Thus, this manuscript addresses a critical challenge and the concept of this work is interesting.

Decomposing ECG signals to obtain different time-frequency components, and using CNN for automatic feature extraction seem well suited for identification of signatures that may be predictive of arrhythmia.

A strength of this work is obviously the innovative automatic ECG classification method that is based on a combination of Continuous Wavelet Transform (CWT) and Convolutional Neural Network (CNN) for ECG classification. It is therefore possible for potential clinical/translational impact, which gives this work particular importance.

The authors employed denoising of the ECG signal, heartbeat segmentation, RR interval extraction, transforming the ECG signal to the time-frequency domain to facilitate feature extraction, and classification based on CNN.

Here are some comments:

1) The original scalogram is 100*200, which authors then resampled it to 100*100 in order to reduce computational cost. It will be interesting to get a comment from authors of the impact of this resampling on classification performance.

2] For the CNN hyper parameter settings, authors used cross-entropy as the loss function, Adam optimizer, He weights initialization [26], a learning rate of 0.001, a batch size 1024 and maximum epoch of 30. Is there any particular reason why authors did not use hyper parameter optimization? Might this not have improved classification performance?

Author Response

Comment 1: The original scalogram is 100*200, which authors then resampled it to 100*100 in order to reduce computational cost. It will be interesting to get a comment from authors of the impact of this resampling on classification performance.

Response: In fact, in our experimental phase, we have tested the performance before and after resampling, and there is no significant difference in performance between the two. This may be because we only sampled the time dimension. The sampling frequency of the original heartbeat is 360Hz. After resampling, the frequency becomes 180Hz. According to the Nyquist criterion, the maximum effective frequency is 90Hz. While the main information of the heartbeat is 0~50Hz. This means that the main information is not lost after resampling.

Comment 2: For the CNN hyper parameter settings, authors used cross-entropy as the loss function, Adam optimizer, He weights initialization [26], a learning rate of 0.001, a batch size 1024 and maximum epoch of 30. Is there any particular reason why authors did not use hyper parameter optimization? Might this not have improved classification performance?

Response: We thank the reviewer for pointing out this issue that needs clarification. The CNN hyperparameters we use are almost all the default settings in Keras (although we use PyTorch to implement the method). These default settings are recommended as a good start in many deep learning studies. In the study, based on these default hyperparameters, we set the parameters of CWT and CNN according to trail-and-error and similar related research. After determining these parameters, we tried to fine-tune these hyperparameters, but found that they could not further improve the classification performance.

This manuscript is a resubmission of an earlier submission. The following is a list of the peer review reports and author responses from that submission.